# Enhancing the Efficiency of a Cybersecurity Operations Center Using Biomimetic Algorithms Empowered by Deep Q-Learning

**DOI:** 10.3390/biomimetics9060307

**Published:** 2024-05-21

**Authors:** Rodrigo Olivares, Omar Salinas, Camilo Ravelo, Ricardo Soto, Broderick Crawford

**Affiliations:** 1Escuela de Ingeniería Informática, Universidad de Valparaíso, Valparaíso 2362905, Chile; camilo.ravelo@postgrado.uv.cl; 2Escuela de Ingeniería y Negocios, Universidad Viña del Mar, Viña del Mar 2572007, Chile; omar.salinas@uvm.cl; 3Escuela de Ingeniería Informática, Pontificia Universidad Católica de Valparaíso, Valparaíso 2362807, Chile; ricardo.soto@pucv.cl (R.S.); broderick.crawford@pucv.cl (B.C.)

**Keywords:** biomimetic optimization algorithm, deep Q-learning, cyber SOC, security information event management

## Abstract

In the complex and dynamic landscape of cyber threats, organizations require sophisticated strategies for managing Cybersecurity Operations Centers and deploying Security Information and Event Management systems. Our study enhances these strategies by integrating the precision of well-known biomimetic optimization algorithms—namely Particle Swarm Optimization, the Bat Algorithm, the Gray Wolf Optimizer, and the Orca Predator Algorithm—with the adaptability of Deep Q-Learning, a reinforcement learning technique that leverages deep neural networks to teach algorithms optimal actions through trial and error in complex environments. This hybrid methodology targets the efficient allocation and deployment of network intrusion detection sensors while balancing cost-effectiveness with essential network security imperatives. Comprehensive computational tests show that versions enhanced with Deep Q-Learning significantly outperform their native counterparts, especially in complex infrastructures. These results highlight the efficacy of integrating metaheuristics with reinforcement learning to tackle complex optimization challenges, underscoring Deep Q-Learning’s potential to boost cybersecurity measures in rapidly evolving threat environments.

## 1. Introduction

In the digital age, the landscape of the contemporary world is increasingly shaped by technological advancements, where threats in the cyber realm pose significant challenges to enterprises. Recognizing these threats necessitates a nuanced understanding of cybersecurity culture, the development of robust cyber-risk management strategies, and the adoption of a proactive, collaborative approach tailored to each organization’s unique context [1,2]. In response, this paper introduces a novel, adaptable cybersecurity risk management framework designed to seamlessly integrate with the evolving threat landscape, leveraging technological progress and aligning with specific organizational needs.

The advancement of optimization techniques, driven by an expansion in scientific knowledge, has led to notable breakthroughs in various fields, including cybersecurity [3]. Artificial intelligence (AI) plays a pivotal role in this evolution, especially through the development of bio-inspired optimization algorithms. These algorithms, inspired by natural processes, have been instrumental in enhancing cyber-risk management strategies by offering innovative solutions and efficiencies [4]. Despite their effectiveness in solving complex problems, these algorithms can encounter limitations, such as stagnation at local optima, which poses a challenge to achieving global optimization [5]. Nevertheless, this challenge also presents an opportunity for a strategic focus on diversification within the search domain, facilitating significant improvements in cyber-risk management efficacy.

Bio-inspired algorithms often struggle to achieve global optimization due to their inherent design, which tends to favor convergence towards local optima based on immediate environmental information. This can lead to the premature acceptance of suboptimal solutions [6,7]. Addressing this issue is crucial and involves promoting a balanced approach to exploration and exploitation, encouraging the exploration of previously uncharted territories and the pursuit of untapped opportunities, thereby enhancing the identification and mitigation of cyber risks [8].

This research proposes a cutting-edge hybrid algorithm that combines metaheuristic algorithms with reinforcement learning to efficiently search for and identify optimal solutions in global optimization tasks. This approach aims to strike a delicate balance between exploration and exploitation, gradually offering more advantageous solutions over time, while avoiding the pitfalls of premature convergence [9]. By leveraging the strengths of bio-inspired algorithms such as Particle Swarm Optimization (PSO), the Bat Algorithm (BAT), the Gray Wolf Optimizer (GWO), and the Orca Predator Algorithm (OPA) for initial detection, and subsequently optimizing the search process with Deep Q-Learning (DQL), this study seeks to address and overcome the challenges of the exploration–exploitation balance and computational complexity, especially in high-dimensional search spaces [10,11].

Enhancing the methodology outlined in [12], this paper extends the integration of bio-inspired algorithms with Deep Q-Learning to optimize the implementation of Cybersecurity Operations Centers (Cyber SOCs). It focuses on a comprehensive risk and requirement evaluation, the establishment of clear objectives, and the creation of a robust technological infrastructure, featuring key tools such as Security Information and Event Management (SIEM) and Network Intrusion Detection Systems (NIDSs) for effective real-time monitoring and threat mitigation [13,14].

Structured to provide a thorough investigation, this paper is organized as follows: Section 2 offers a detailed review of recent integrations of machine learning with metaheuristics in cybersecurity, highlighting multi-objective optimization challenges. Section 3 delves into preliminary concepts of bio-inspired algorithms, emphasizing the principles of PSO, BAT, GWO, and OPA, alongside a formal introduction to DQL and advancements in Cyber SOC and SIEM technologies. Section 4 outlines the development of the proposed solution, with Section 5 detailing the experimental design methodology. Section 6 analyzes the results, discussing the hybridization’s effectiveness in generating efficient solutions. Finally, Section 7 concludes the study, summarizing key findings and suggesting directions for future research.

## 2. Related Work

The rising frequency and severity of cyberattacks underscore the essential role of cybersecurity in protecting organizational assets. Research such as the study by [15] introduces a groundbreaking multi-objective optimization approach for cybersecurity countermeasures using Genetic Algorithms. Their methodology aims to fine-tune Artificial Immune System parameters to achieve an ideal balance between minimizing risk and optimizing execution time. The robustness of the model is demonstrated through comprehensive testing across a broad spectrum of inputs, showcasing its capacity for a swift and effective cybersecurity response.

In the realm of machine learning (ML), techniques are being increasingly applied across diverse domains, including the creation of advanced machine learning models, enhancing physics simulations, and tackling complex linear programming challenges. The research conducted by [16] delves into the significant impact of machine learning on the domain knowledge of metaheuristics, leading to enhanced problem-solving methodologies. Furthermore, the integration of machine learning with metaheuristics, as explored in studies [17,18], opens up promising avenues for cyber-risk management, showcasing the transformative potential of ML in developing new strategies and enhancing existing cybersecurity mitigation efforts.

The synergy between advanced machine learning techniques and metaheuristics is pivotal in crafting solutions that effectively address the sophisticated and ever-evolving landscape of cyber threats. Notably, research such as [19] emphasizes the utility of integrating Q-Learning with Particle Swarm Optimization for the resolution of combinatorial problems, marking a significant advancement over traditional PSO methodologies. The approach not only enhances solution quality but also exemplifies the effectiveness of learning-based hybridizations in the broader context of swarm intelligence algorithms, providing a novel and adaptable methodology for tackling optimization challenges.

Innovative algorithmic design further underscores the progress in optimization techniques, with the introduction of the self-adaptive virus optimization algorithm by [20]. The novel algorithm improves upon the conventional virus optimization algorithm by minimizing the reliance on user-defined parameters, thus facilitating a broader application across various problem domains. The dynamic adaptation of its parameters significantly elevates the algorithm’s performance on benchmark functions, showcasing its superiority, particularly in scenarios where the traditional algorithm exhibited limitations. The advancement is achieved by streamlining the algorithm, reducing controllable parameters to a singular one, thereby enhancing its efficiency and versatility for continuous domain optimization challenges.

The discourse on metaheuristic algorithms for solving complex optimization problems is enriched by [21], which addresses the manual design of these algorithms without a cohesive framework. Proposing a General Search Framework to amalgamate diverse metaheuristic strategies, the method introduces a systematic approach for the selection of algorithmic components, facilitating the automated design of sophisticated algorithms. The framework enables the development of novel, population-based algorithms through reinforcement learning, marking a pivotal step towards the automation of algorithm design supported by effective machine learning techniques.

In the domain of intrusion detection, Ref. [22] introduces an innovative technique, a metaheuristic with a deep learning-enabled intrusion detection system for a secured smart environment (MDLIDS–SSE), which combines metaheuristics with deep learning to secure intelligent environments. Employing Z-score normalization for data preprocessing via the improved arithmetic optimization algorithm-based feature selection (IAOA–FS), the method achieves high precision in intrusion classification, surpassing recent methodologies. Experimental validation underscores its potential in safeguarding smart cities, buildings, and healthcare systems, demonstrating promising results in accuracy, recall, and detection rates.

Additionally, the Q-Learning Vegetation Evolution algorithm, as presented in [23], exemplifies the integration of Q-Learning for optimizing coverage in numerical and wireless sensor networks. The approach, featuring a mix of exploitation and exploration strategies and the use of online Q-Learning for dynamic adaptation, demonstrates significant improvements over conventional methods through rigorous testing on CEC2020 benchmark functions and real-world engineering challenges. The research contributes a sophisticated approach to solving complex optimization problems, highlighting the efficacy of hybrid strategies in the field.

In the sphere of cyber-risk management, particularly from the perspective of the Cyber SOC and SIEM, research efforts focus on strategic optimization, automated responses, and adaptive methodologies to navigate the dynamic cyber-threat landscape. Works such as [12,24] explore efficient strategies for designing network topologies and optimizing cybersecurity incident responses within SIEM systems. These studies leverage multi-objective optimization approaches and advanced machine learning models, like Deep Q neural networks, to enhance decision-making processes, showcasing significant advancements in the automation and efficiency of cybersecurity responses.

Emerging strategies in intrusion detection and network security, highlighted by [25,26], emphasize the integration of reinforcement learning with oversampling and undersampling algorithms, and the combination of Particle Swarm Optimization–Genetic Algorithm with the LSTM–GRU of deep learning that fused the GRU (gated recurrent unit) and LSTM (long short-term memory). These approaches demonstrate a significant leap forward in detecting various types of attacks within Internet of Things (IoT) networks, showcasing the power of combining machine learning and optimization techniques for IoT security. The model’s accuracy in classifying different attack types, as tested on the CICIDS-2017 dataset, outperforms existing methods and suggests a promising direction for future research in this domain.

Furthermore, Ref. [27] introduces a semi-supervised alert filtering scheme that leverages semi-supervised learning and clustering techniques to efficiently distinguish between false and true alerts in network security monitoring. The method’s effectiveness, as evidenced by its superior performance over traditional approaches, offers a fresh perspective on alert filtering, significantly contributing to the improvement of network security management by reducing alert fatigue.

The exploration of machine learning’s effectiveness and cost-efficiency in NIDS for small and medium enterprises (SMEs) in the UK is presented in [28]. The study assesses various intrusion detection and prevention devices, focusing on their ability to manage zero-day attacks and related costs. The research, conducted during the COVID-19 pandemic, investigates both commercial and open-source NIDS solutions, highlighting the balance between cost, required expertise, and the effectiveness of machine learning-enhanced NIDS in safeguarding SMEs against cyber threats.

From the perspective of Cyber SOCs, Ref. [29] addresses the increasing complexity of cyberattacks and their implications for public sector organizations. The study proposes a ‘Wide–Scope CyberSOC’ model as a unique outsourced solution to enhance cybersecurity awareness and implementation across various operational domains, tackling the challenges faced by public institutions in building a skilled cybersecurity team and managing the blend of internal and external teams amidst the prevailing outsourcing trend.

Lastly, Ref. [30] offers a comprehensive analysis of the bio-inspired Internet of Things, underscoring the synergy between biomimetics and advanced technologies. The research evaluates the current state of Bio-IoT, focusing on its benefits, challenges, and future potential. The integration of natural principles with IoT technology promises to create more efficient and adaptable solutions, addressing key challenges such as data security and privacy, interoperability, scalability, energy management, and data handling.

## 3. Preliminaries

In this study, we integrated bio-inspired algorithms with an advanced machine learning technique to tackle a complex optimization problem. Specifically, we utilized Particle Swarm Optimization, the Bat Algorithm, the Grey Wolf Optimizer and the Orca Predator Algorithm, which are inspired by the intricate processes and behaviors observed in nature and among various animal species. These algorithms were improved by incorporating reinforcement learning through Deep Q-Learning into the search process of bio-inspired methods.

### 3.1. Particle Swarm Optimization

Particle Swarm Optimization is a computational method that simulates the social behavior observed in nature, such as birds flocking or fish schooling, to solve optimization problems [31]. This technique is grounded in the concept of collective intelligence, where simple agents interact locally with one another and with their environment to produce complex global behaviors.

In PSO, a swarm of particles moves through the solution space of an optimization problem, with each particle representing a potential solution. The movement of these particles is guided by their own best-known positions in the space, as well as the overall best-known positions discovered by any particle in the swarm. This mechanism encourages both individual exploration of the search space and social learning from the success of other particles. The position of each particle is updated according to Equations (Equation 1) and (Equation 2):(1)$$v_{i}\left(t+1\right)=wv_{i}\left(t\right)+c_{1}rand_{1}\left(pbest_{i}-x_{i}\left(t\right)\right)+c_{2}rand_{2}\left(gbest-x_{i}\left(t\right)\right)$$
(2)$$x_{i}\left(t+1\right)=x_{i}\left(t\right)+v_{i}\left(t+1\right)$$
where $v_{i}\left(t+1\right)$ is the velocity of particle *i* at iteration t+1. *w* is the weight of inertia that helps balance exploration and exploitation. $c_{1}$ and $c_{1}$ are coefficients representing self-confidence and social trust, respectively. $rand_{1}$ and $rand_{1}$ are random numbers between 0 and 1. $pbest_{i}$ is the best known position for the particle *i* and gbest is the best position known of the entire population. Finally, $x_{i}\left(t\right)$ and $x_{i}\left(t+1\right)$ represent the current position of the particle *i* and the next one, respectively.

The algorithm iterates these updates, allowing particles to explore the solution space, with the aim of converging towards the global optimum. The parameters *w*, $c_{1}$, and $c_{2}$ play crucial roles in the behavior of the swarm, affecting the convergence speed and the algorithm’s ability to escape local optima.

PSO is extensively employed due to its simplicity, efficiency, and versatility, enabling its application across a broad spectrum of optimization problems. Its ability to discover solutions without requiring gradient information renders it especially valuable for problems characterized by complex, nonlinear, or discontinuous objective functions.

### 3.2. Bat Algorithm

The Bat Algorithm is an optimization technique inspired by the echolocation behavior of bats. It simulates the natural echolocation mechanism that bats use for navigation and foraging. This algorithm captures the essence of bats’ sophisticated biological sonar systems, translating the dynamics of echolocation and flight into a computational algorithm capable of searching for global optima in complex optimization problems [6].

In the Bat Algorithm, a population of virtual bats navigates the solution space, where each bat represents a potential solution. The bats use a combination of echolocation and a random walk to explore and exploit the solution space effectively. They adjust their echolocation parameters, such as frequency, pulse rate, and loudness, to locate prey, analogous to finding the optimal solutions in a given problem space. The algorithm employs the Equations (Equation 3)–(Equation 5), for updating the bats’ positions and velocities:(3)$$f_{i}=f_{min}+\left(f_{max}-f_{min}\right)\beta$$
(4)$$v_{i}\left(t+1\right)=v_{i}\left(t\right)+\left(x_{i}\left(t\right)-gbest\right)f_{i}$$
(5)$$x_{i}\left(t+1\right)=x_{i}\left(t\right)+v_{i}\left(t+1\right)$$
where $f_{i}$ is the frequency of the bat *i*, ranging from $f_{min}$ to $f_{max}$ with β being a random number between 0 and 1. $v_{i}\left(t+1\right)$ represents the velocity of bat *i* at iteration t+1, and gbest signifies the global best solution found by any bat. $x_{i}\left(t+1\right)$ denotes the position of bat *i* for the next iteration.

Additionally, to model the bats’ local search and exploitation capability, a random walk is incorporated into the best solution found so far (see Equation (Equation 6)). This is achieved by modifying a bat’s position using the average loudness *A* of all the bats and the pulse emission rate *r*, guiding the search towards the optimum:(6)$$x_{new}=x_{gbest}+\epsilon A$$
where $x_{new}$ represents a new solution generated by local search around the global best position $x_{gbest}$, and ϵ is a random number drawn from a uniform distribution. The values of *A* and *r* decrease and increase, respectively, over the course of iterations, fine-tuning the balance between exploration and exploitation based on the proximity to the prey, i.e., the optimal solution.

The Bat Algorithm’s efficiency stems from its dual approach of global search, facilitated by echolocation-inspired movement and local search, enhanced by the random walk based on pulse rate and loudness. This combination allows the algorithm to explore vast areas of the search space while also intensively searching areas near the current best solutions.

### 3.3. Gray Wolf Optimizer

The Gray Wolf Optimizer is an optimization algorithm inspired by the social hierarchy and hunting behavior of gray wolves in nature. This algorithm mimics the leadership and team dynamics of wolves in packs to identify and converge on optimal solutions in multidimensional search spaces [32]. The core concept behind GWO is the emulation of the way gray wolves organize themselves into a social hierarchy and collaborate during hunting, applying these behaviors to solve optimization problems.

In a gray wolf pack, there are four types of wolves: alpha (α), beta (β), delta (δ), and omega (ω), representing the leadership hierarchy. The alpha wolves lead the pack, followed by beta and delta wolves, with omega wolves being at the bottom of the hierarchy. This social structure is translated into the algorithm where the best solution is considered the alpha, the second-best the beta, and the third-best the delta. The rest of the candidate solutions are considered omega wolves, and they follow the lead of the alpha, beta, and delta wolves towards the prey (optimal solution).

The positions of the wolves are updated based on the positions of the alpha, beta, and delta wolves, simulating the hunting strategy and encircling of prey. The mathematical models for updating the positions of the gray wolves are given by Equations (Equation 7) and (Equation 8):(7)$$\vec{D}=\left|\vec{C}\cdot {\vec{x}}_{p}\left(t\right)-\vec{x}\left(t\right)\right|$$
(8)$$\vec{x}\left(t+1\right)={\vec{x}}_{p}\left(t\right)-\vec{A}\cdot \vec{D}$$
where ${\vec{x}}_{p}\left(t\right)$ represents the position vector of the prey (or the best solution found so far), $\vec{x}$ is the position vector of a wolf, $\vec{A}$ and $\vec{C}$ are coefficient vectors, and *t* indicates the current iteration. The vectors $\vec{A}$ and $\vec{C}$ are calculated by Equations (Equation 8) and (Equation 10):(9)$$\vec{A}=2\cdot \vec{a}\cdot {\vec{r}}_{1}-\vec{a}$$
(10)$$\vec{C}=2\cdot {\vec{r}}_{2}$$
where $\vec{a}$ linearly decreases from 2 to 0 over the course of iterations, and ${\vec{r}}_{1}$, ${\vec{r}}_{2}$ are random vectors in [0,1].

The hunting (optimization) is guided mainly by the alpha, beta, and delta wolves, with omega wolves following their lead. The algorithm effectively simulates the wolves’ approach and encircling of prey, exploration of the search area, and exploitation of promising solutions.

### 3.4. Orca Predator Algorithm

The Orca Predator Algorithm draws inspiration from the sophisticated hunting techniques of orcas, known for their strategic and cooperative behaviors [33]. Orca societies are characterized by complex structures and collaborative efforts in predation, employing echolocation for navigation and prey detection in their aquatic environments. OPA models solutions as *n*-dimensional vectors within a solution space $\vec{x}={\left[x_{1},x_{2},\ldots ,x_{n}\right]}^{T}$, mimicking these marine predators’ approaches to tracking and capturing prey.

OPA’s methodology encompasses two main phases reflective of orca predation: the chase, involving herding and encircling tactics, and the attack, focusing on the actual capture of prey. During the chase phase, the algorithm alternates between herding prey towards the surface and encircling it to limit escape opportunities, with the decision based on a parameter *p* and a random number *r* within [0,1]. The attack phase simulates the final assault on the prey, highlighting the importance of coordination and precision.
(11)$${\vec{x}}_{chase,1,i}^{t}=a\times \left(d\times {\vec{x}}_{best}^{t}-F\times \left(b\times M^{t}+c\times {\vec{x}}_{i}^{t}\right)\right)$$
(12)$${\vec{x}}_{chase,2,i}^{t}=e\times {\vec{x}}_{best}^{t}-{\vec{x}}_{i}^{t}$$
(13)$$M=\frac{\sum_{i=1}^{N}{\vec{x}}_{i}^{t}}{N},\;c=1-b$$
(14)$${\vec{x}}_{new}=\left\{\begin{array}{cc} {\vec{x}}_{chase,1,i}^{t}={\vec{x}}_{i}^{t}+{\vec{x}}_{chase,1,i}^{t} & \mathrm{when}\;q>rand\\ {\vec{x}}_{chase,2,i}^{t}={\vec{x}}_{i}^{t}+{\vec{x}}_{chase,2,i}^{t} & \mathrm{when}\;q\leq rand \end{array}\right.$$

Equations (Equation 11)–(Equation 14) detail the algorithm’s dynamics, modeling velocity and spatial adjustments reflective of orca hunting behaviors. ${\vec{x}}_{i}^{t}$ represents the position of the *i*-th orca at time *t*, with ${\vec{x}}_{best}^{t}$ denoting the optimal solution’s position. Parameters *a*, *b*, *d*, and *e* are random coefficients that influence the algorithm’s exploration and exploitation mechanisms, with *F* indicating the attraction force between agents.

After herding prey to the surface, orcas coordinate to finalize the hunt, using their positions and the positions of randomly chosen peers to strategize their attack. This collective behavior is encapsulated in Equations (Equation 15) and (Equation 16), illustrating the algorithm’s mimicry of orca hunting techniques:(15)$$x_{chase,3,i,k}^{t}=x_{j_{1},k}^{t}+u\times \left(x_{j_{2},k}^{t}-x_{j_{3},k}^{t}\right)$$
(16)$$u=2\times \left(rand-0.5\right)\times \frac{MaxIter-t}{MaxIter}$$

These formulations demonstrate how orcas adapt their positions based on the dynamics of their surroundings and the behaviors of their pod members, optimizing their strategies to efficiently capture prey. Through this algorithm, the intricate and collaborative nature of orca predation is leveraged as a metaphor for solving complex optimization problems, with a focus on enhancing solution accuracy and efficiency.

### 3.5. Reinforcement Learning

Reinforcement learning revolves around the concept of agents operating autonomously to optimize rewards through their decisions, as outlined in comprehensive studies [34]. These agents navigate their learning journey via a trial and error mechanism, pinpointing behaviors that accrue maximum rewards, both immediately and in the future, a hallmark trait of reinforcement learning [35].

During the reinforcement learning journey, agents are in constant interaction with their surroundings, engaging with essential elements like the policy, value function, and, at times, a simulated representation of the environment [36,37,38,39]. The value function assesses the potential success of the actions taken by the agent within its environment, while adjustments in the agent’s policy are influenced by the rewards received.

One pivotal reinforcement learning method, Q-Learning, aims to define a function that evaluates the potential success of an action $a_{t}$ in a certain state $s_{t}$ at time *t* [40,41]. This evaluation function, or Q function, undergoes updates as per Equation (Equation 17):(17)$$Q\left(s_{t},a_{t}\right)\leftarrow Q\left(s_{t},a_{t}\right)+\alpha \left[r_{t+1}+\gamma \max_{a}Q\left(s_{t+1},a\right)-Q\left(s_{t},a_{t}\right)\right]$$

Here, α symbolizes the learning rate, and γ represents the discount factor, with $r_{t+1}$ being the reward after executing action $a_{t}$.

Deep Q-Learning (DQL) merges the robust capabilities of deep learning with the adaptive mechanisms of reinforcement learning, offering significant advancements in handling complex, high-dimensional environments [42]. This integration allows agents to learn and refine strategies through direct interaction with their environment, optimizing their decision-making processes over time. DQL has proven particularly effective in applications such as autonomous vehicles, robotics, and complex game environments, where agents must make real-time decisions based on incomplete information. Despite its strengths, DQL faces challenges such as the need for extensive training data and potential overfitting, underscoring the importance of ongoing research to enhance its stability and efficiency in diverse applications.

Within Deep Q-Learning, the Q function is articulated as $Q(s_{t},a_{t};\theta )$, where $s_{t}$ denotes the present state, $a_{t}$ the action undertaken by the agent at time *t*, and θ the network’s weights [43]. The Q function’s update mechanism is guided by Equation (Equation 18):(18)$$Q\left(s_{t},a_{t};\theta \right)\leftarrow Q\left(s_{t},a_{t};\theta \right)+\alpha \left[r_{t+1}+\gamma \max_{a_{t+1}}Q\left(s_{t+1},a_{t+1};\theta^{-}\right)-Q\left(s_{t},a_{t};\theta \right)\right]$$

Here, $s_{t+1}$ and $a_{t+1}$ indicate the subsequent state and action at time t+1, respectively. The learning rate α influences the extent of Q value function updates at each learning step. A higher α facilitates rapid adjustment to environmental changes, beneficial during the learning phase’s early stages or in highly variable settings. Conversely, a lower α ensures a more gradual and steady learning curve but might extend the convergence period. The discount factor γ prioritizes the future over immediate rewards, promoting strategies focused on long-term gain. In contrast, a lower γ favors immediate rewards, suitable for less predictable futures or scenarios necessitating quick policy development. The reward $r_{t+1}$ is received post-action $a_{t}$ execution in state $s_{t}$, with $\theta^{-}$ denoting the parameters of a secondary neural network that periodically synchronizes with θ to enhance training stability.

A hallmark of Deep Q-Learning is the incorporation of replay memory, a pivotal component of its learning framework [44,45]. Replay memory archives the agent’s experiences as tuples $\langle s_{t},a_{t},r_{t+1},s_{t+1}\rangle$, with each tuple capturing a distinct experience involving the current state $s_{t}$, the executed action $a_{t}$, the obtained reward $r_{t+1}$, and the ensuing state $s_{t+1}$. This methodology of preserving and revisiting past experiences significantly improves learning efficiency and efficacy, enabling the agent to draw from a broader spectrum of experiences. It also diminishes the sequential dependency of learning events, a crucial strategy for mitigating the risk of over-reliance on recent data and fostering a more expansive learning approach. Furthermore, DQL employs the mini-batch strategy for extracting experiences from replay memory throughout the training phase [46]. Rather than progressing from individual experiences one by one, the algorithm opts for a random selection of mini-batches of experiences. This technique of batch sampling bolsters learning stability by ensuring sample independence and optimizes computational resource utilization.

DQL has proven highly effective across a range of applications, demonstrating its versatility and robust decision-making capabilities [47,48]. In autonomous vehicles [49], DQL developed systems that process vast sensory data to make real-time navigational decisions, significantly enhancing road safety and efficiency. In the realm of robotics, an application of a Deep Q-Learning algorithm to enhance the positioning accuracy of an industrial robot was studied in [50]. Additionally, DQL’s strategic decision-making prowess was applied in healthcare to optimize the security and privacy of healthcare data in IoT systems, focusing on authentication, malware, and DDoS attack mitigation, and evaluating performance through metrics like energy consumption and accuracy [51]. In the financial sector, the study [52] introduced an automated trading system that combines reinforcement learning with a deep neural network to predict share quantities and employs transfer learning to overcome data limitations. This approach significantly boosts profits across various stock indices, outperforming traditional systems in volatile markets. Moreover, in supply chain and logistics, the manuscript [53] reviewed the increasing deep reinforcement learning to address challenges stemming from evolving business operations and E-commerce growth, discussing methodologies, applications, and future research directions.

DQL is governed by a loss function according to Equation (Equation 19), which measures the discrepancy between the estimated Q and target values:(19)$$\mathrm{Loss}\left(\theta_{t}\right)=E\times \left[{\left(y-Q(s_{t},a_{t};\theta )\right)}^{2}\right]$$
where *y* is the target value, calculated by Equation (Equation 20):(20)$$y=r_{t+1}+\gamma \times \max_{a_{t+1}}Q\left(s_{t+1},a_{t+1};\theta^{-}\right)$$

Here, $r_{t+1}$ is the reward received after taking action $a_{t}$ in the state $s_{t}$, and γ is the discount factor, which balances the importance of short-term and long-term rewards. The formulation $\max_{a^{\prime }}Q\left(s_{t+1},a_{t+1};\theta^{-}\right)$ represents the maximum estimated value for the next state $s_{t+1}$, according to the target network with parameters $\theta^{-}$. $Q(s_{t},a_{t};\theta_{t})$ is the Q value estimated by the evaluation network for the current state $s_{t}$ and action $a_{t}$, using the current parameters $\theta_{t}$. In each training step in DQL, the evaluation network receives a loss function, which is backpropagated based on a batch of experiences randomly selected from the experience replay memory. The evaluation network’s parameter, θ, is then updated by minimizing the loss function through the Stochastic Gradient Descent (SGD) function. After several steps, the target network’s parameter, $\theta^{-}$, is updated by assigning the latest parameter θ to $\theta^{-}$. After a period of training, the two neural networks are trained stably.

### 3.6. Cybersecurity Operations Centers

Recent years have seen many organizations establish Cyber SOCs in response to escalating security concerns, necessitating substantial investments in technology and complex setup processes [54]. These centralized hubs enhance incident detection, investigation, and response capabilities by analyzing data from various sources, thereby increasing organizational situational awareness and improving security issue management [55]. The proliferation of the Internet and its integral role in organizations brings heightened security risks, emphasizing the need for continuous monitoring and the implementation of optimization methods to tackle challenges like intrusion detection and prevention effectively [56].

Security Information and Event Management systems have become essential for Cyber SOCs, playing a critical role in safeguarding the IT infrastructure by enhancing cyber-threat detection and response, thereby improving operational efficiency and mitigating security incident impacts [57]. The efficient allocation of centralized NIDS sensors through an SIEM system is crucial for optimizing detection coverage and operational efficiency, considering the organization’s specific security needs [58]. This strategic approach allows for cohesive management and comprehensive security data analysis, leading to a faster and more effective response to security incidents [59]. SIEM systems, widely deployed to manage cyber risks, have evolved into comprehensive solutions that offer broad visibility into high-risk areas, focusing on proactive mitigation strategies to reduce incident response costs and times [60]. Figure 1 illustrates the functional characteristics of a Cyber SOC.

Today’s computer systems are universally vulnerable to cyberattacks, necessitating continuous and comprehensive security measures to mitigate risks [61]. Modern technology infrastructures incorporate various security components, including firewalls, intrusion detection and prevention systems, and security software on devices, to fortify against threats [62]. However, the autonomous operation of these measures requires the integration and analysis of data from different security elements for a complete threat overview, highlighting the importance of Security Information and Event Management systems [63]. As the core of Cyber SOCs, SIEM systems aggregate data from diverse sources, enabling effective threat management and security reporting [64].

SIEM architectures consist of key components such as source device integration, log collection, and event monitoring, with a central engine performing log analysis, filtering, and alert generation [65,66]. These elements work together to provide real-time insights into network activities, as depicted in Figure 2.

NIDS sensors, often based on cost-effective Raspberry Pi units, serve as adaptable and scalable modules for network security, requiring dual Ethernet ports for effective integration into the SIEM ecosystem [67]. This study aims to enhance the assignment and management of NIDS sensors within a centralized network via SIEM, improving the optimization of sensor deployment through the application of Deep Q-Learning to metaheuristics, advancing upon previous work [12].

In this context, cybersecurity risk management is essential for organizations to navigate the evolving threat landscape and implement appropriate controls [68]. It aims to balance securing networks and minimizing losses from vulnerabilities [69], requiring continuous model updates and the strategic deployment of security measures [70]. Cyber-risk management strategies, including the adoption of SIEM systems, are vital for monitoring security events and managing incidents [69].

The optimization problem focuses on deploying NIDS sensors effectively, considering cost, benefits, and indirect costs of non-installation. This involves formulations to minimize sensor costs (Equation (Equation 22)), maximize benefits (Equation (Equation 23)), and minimize indirect costs (Equation (Equation 24)), with constraints ensuring sufficient sensor coverage (Equation (Equation 25)) and network reliability (Equation (Equation 26)):(21)$$F\left(\vec{x}\right)=\langle f_{1}\left(\vec{x}\right),f_{2}\left(\vec{x}\right),f_{3}\left(\vec{x}\right)\rangle$$
(22)$$f_{1}\left(\vec{x}\right):\min_{x_{ij}\in X}\sum_{i=1}^{s}\sum_{j=1}^{n}x_{ij}c_{ij}$$
(23)$$f_{2}\left(\vec{x}\right):\max_{x_{ij}\in X}\sum_{j=1}^{n}x_{ij}d_{ij},\;\forall i$$
(24)$$f_{3}\left(\vec{x}\right):\min_{x_{ij}\in X}\sum_{j=1}^{n}\left(1-x_{ij}\right)i_{ij},\;\forall i$$
(25)$$\sum_{j=1}^{n}x_{ij}\geq \;1,\;\forall i$$
(26)$$\frac{\sum_{j=1}^{n}pj\left(1-xij\right)}{\sum_{j=1}^{n}p_{j}}\leq \left(1-u\right),\;\forall i$$

This streamlined approach extends the model to larger networks and emphasizes the importance of regular updates and expert collaboration to improve cybersecurity outcomes [12,71].

Expanding on research [12] which optimized NIDS sensor allocation in medium-sized networks, this study extends the approach to larger networks. By analyzing a case study, this research first tackled instance zero with ten VLANs, assigning qualitative variables to each based on operational importance and failure susceptibility for strategic NIDS sensor placement. This formulation led to an efficient allocation of NIDS sensors for the foundational instance zero, as depicted in Figure 3. The study scaled up to forty additional instances, providing a robust examination of NIDS sensor deployment strategies in varied network configurations.

## 4. Developed Solution

This solution advances the integration of bio-inspired algorithms—Particle Swarm Optimization, the Bat Algorithm, the Grey Wolf Optimizer, and the Orca Predator Algorithm—with Deep Q-Learning to dynamically fine-tune the parameters of these algorithms. Utilizing the collective and adaptive behaviors of PSO, BAT, GWO, and OPA alongside the capabilities of DQL to handle extensive state and action spaces, we enhanced the efficiency of feature selection. Inspired by the natural strategies of their respective biological counterparts and combined with DQL’s proficiency in managing high-dimensional challenges [33,72], this approach innovates optimization tactics while effectively addressing complex combinatorial issues.

DQL is pivotal for shifting towards exploitation, particularly in later optimization phases. As PSO, BAT, GWO, and OPA explore the solution space, DQL focuses the exploration on the most promising regions through an epsilon-greedy policy, optimizing action selection as the algorithm progresses and learns [73].

Each algorithm functions as a metaheuristic with agents (particles, bats, wolves, or agents) representing search agents within the binary vector solution space.

DQL’s reinforcement learning strategy fine-tunes the operational parameters of these algorithms, learning from their performance outcomes to enhance exploration and exploitation balance. Through replay memory, DQL benefits from historical data, incrementally improving NIDS sensor mapping for an SIEM system.

Figure 4 displays the collaborative workflow between the bio-inspired algorithms and DQL, showcasing an efficient and effective optimization methodology that merges nature-inspired exploration with DQL’s adaptive learning.

The essence of our methodology is captured in the pseudocode of Algorithm 1, beginning with dataset input and leading to the global best solution identification. This process involves initializing agents, adjusting their positions and velocities, and employing a training phase to compute fitness, followed by DQL’s refinement of exploration and exploitation strategies.

The core loop iterates until reaching a specified limit, with each agent’s position and velocity updated and fitness evaluated for refining the search strategy.

Finally, the computational complexity of our metaheuristic is O(kn), where *n* is the problem dimension and *k* is the number of iterations or population size, reflecting total function evaluations during the algorithm’s execution. The complexity of the Deep Q-Learning (DQL) algorithm is typically O(MN) [74,75], with *M* as the sample size and *N* as the number of network parameters, crucial for comprehensive dataset analysis. Although *M* and *N* can be large, their fixed nature justifies the increased computational demand for enhanced results. Furthermore, ongoing advancements in computing technology help offset the impact of this increased complexity.
**Algorithm 1:** Enhanced bio-inspired optimization method
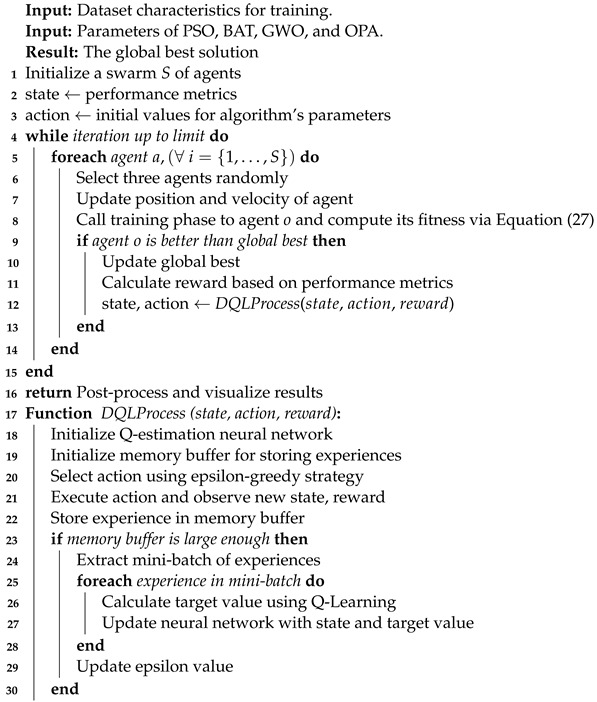


## 5. Experimental Setup

Forty instances were proposed for the experimental stage. These instances entailed random operating parameters, which are detailed below for each instance: the number of operational VLANs, the types of sensors used, the range of sensor costs, the range of benefits associated with sensor installation in a particular VLAN, the range of indirect costs incurred when a sensor is not installed in a given VLAN, and the probability of non-operation for a given VLAN.

Additionally, it is important to note that, as per the mathematical modeling formulation outlined earlier, one of its constraints mandates a minimum operational or uptime availability of ninety percent for the organization’s network. The specific values for each instance are provided in detail in Table 1.

Once the solution vector has been altered, it becomes necessary to implement a binarization step for the usage of continuous metaheuristics in a binary domain [76]. This involves comparing the sigmoid function to a randomly uniform value δ that falls within the range of 0 and 1. Subsequently, a conversion function, for instance, $[1/\left(1+e^{-x_{i}^{j}}\right)]>\delta$, is employed as a method of discretization. In this scenario, if the statement holds true, then $x_{i}^{j}\leftarrow 1$. Conversely, if it does not hold true, then $x_{i}^{j}\leftarrow 0$.

Our objective was to devise plans and offer recommendations for the trial phase, thereby demonstrating that the recommended strategy is a feasible solution for determining the location of the NIDS sensor. The time taken to solve was calculated to gauge the duration of metaheuristics required to achieve efficient solutions. We used the highest value as a critical measure to evaluate subsequent outcomes, which the Equation determines (Equation 27).
(27)$$\begin{array}{c} \sum_{{\left(p,q\right)}_{p\neq q}\;\in \;K\;}\underset{max}{\underset{︸}{\frac{f_{p}\left(\vec{x}\right)}{e_{p}\left({\vec{x}}^{best}\right)}\omega_{p}}}+\underset{min}{\underset{︸}{\frac{\hat{c}-f_{q}\left(\vec{x}\right)}{\hat{c}-e_{q}\left({\vec{x}}^{best}\right)}\omega_{q}}},\;\omega_{\left(p,q\right)}⩾0 \end{array}$$
where $\omega_{\left(p,q\right)}$ represents weight of objective functions and $\sum \omega_{\left(p,q\right)}=1$ must be satisfied. Values of $\omega_{\left(p,q\right)}$ are defined by analogous estimating. $f_{\left(p,q\right)}\left(\vec{x}\right)$ is the single-objective function and $e_{\left(p,q\right)}\left({\vec{x}}^{best}\right)$ stores the best value met independently. Finally, $\hat{c}$ is an upper bound of minimization single-objective functions.

Following this, we employed ordinal examination to assess the adequacy of the strategy. Subsequently, we elaborate on the hardware and software utilized to duplicate computational experiments. Outcomes are depicted through tables and graphics.

We highlight that test scenarios were developed using standard simulated networks designed to mimic the behavior and characteristics of real networks. These simulations represent the operational characteristics of networks in organizations of various sizes, from minor to medium and large. Depending on its scale and extent, defined by the number of VLANs, each VLAN consists of multiple devices, such as computers, switches, and server farms, along with their related connections. The research evaluated test networks that varied in size, starting from networks with ten VLANs, moving to networks with twenty-five VLANs, and extending to more extensive networks with up to fifty VLANs. The simulation considered limitations such as bandwidth capacity, latency, packet loss, and network congestion by replicating the test networks and considering their functional and working properties. These aspects, along with other factors, were critical in defining the uptime of each VLAN. Network availability is defined as the time or percentage during which the network remains operational and accessible, without experiencing significant downtime. For this study, it was essential that networks maintained a minimum availability of 90%, as interruptions and periods of downtime may occur due to equipment failure, network congestion, or connectivity problems. Implementing proactive monitoring through the SIEM will ensure high availability on the network.

### 5.1. Methodology

In this study, we employed a multi-phase research approach to ensure thorough analysis and robust findings. Initially, we identified the key variables influencing our research question through an extensive literature review. Following this, we designed our experimental setup to test these variables under controlled conditions, detailing the participant selection criteria, data collection methods, and the statistical tools used for analysis. This meticulous approach enabled us to confirm our data’s reliability and guarantee that our results could be replicated and validated by future studies.

The forty instances representing networks of various sizes and complexities, described in Table 1, were used to evaluate the performance between the native and enhanced metaheuristics following the principles established in [77]. The methodological proposal consisted of an analytical comparison between the hybridization results, that is, the results obtained from the original form of the algorithm and the results obtained with the application of Deep Q-Learning. To achieve this, we implemented the following methodological approach:Preparation and planning: In this phase, network instances that emulated real-world cases, from medium-sized networks to large networks, were generated, randomly covering the various operational and functional scenarios of modern networks. Subsequently, the objectives to achieve were defined as having a secure, operational, and highly available network. These objectives were to minimize the number of NIDS sensors assigned to the network, maximize the installation benefits, and minimize the indirect costs of non-installation. Experiments were designed to systematically evaluate hybridization improvements in a controlled manner, ensuring balanced optimization of the criteria described above.Execution and assessment: We carried out a comprehensive evaluation of both native and improved metaheuristics, analyzing the quality of the solutions obtained and the efficiency in terms of calculation and convergence characteristics. We implement comprehensive tests to perform performance comparisons with descriptive statistical methods and performed the Mann–Whitney–Wilcoxon test for comparative analysis. This method involves determining the appropriateness of each execution for each given instance.Analysis and validation: We performed a comprehensive and in-depth analysis to understand the influence of Deep Q-Learning and the behavior of the PSO, BAT, GWO, and OPA metaheuristics in generating efficient solutions for the corresponding instances. To do this, comparative tables and graphs of the solutions generated by the native and improved metaheuristics were built.

### 5.2. Implementation Aspects

To ensure clarity and transparency of our experimental design, we present our experimental parameters and platforms in Table 2, below. This table outlines all crucial aspects of our setup, allowing for the easy replication of our methods and verification of our results.

## 6. Results and Discussion

Table 3, Table 4, Table 5, Table 6 and Table 7 shows the main findings corresponding to the execution of the native metaheuristics and the metaheuristics improved with Deep Q-Learning. The tables are structured into forty sections (one per instance), each consisting of six rows that statistically describe the value of the metric corresponding to the scalarization of objectives, considering the best value obtained as the minimum value and the worst value obtained as the maximum value. The median represents the middle value, and the mean denotes the average of the results, while the standard deviation (STD) and the interquartile range (IQR) quantify the variability in the findings. Concerning columnar representation, PSO, BAT, GWO, and OPA detail results for bio-inspired optimizers lacking a learning component. PSODQL, BATDQL, GWODQL, and OPADQL represent our enhanced versions of biomimetic algorithms.

When analyzing instances one to nine, it is evident that both the native metaheuristics and the metaheuristics improved with Deep Q-Learning produce identical solutions and metrics, given the low complexity of the IT infrastructure of these first instances; however, despite generating exact values regarding the best scalarization value, instances six, eight, and nine show variations in their generation, which can be seen in the variation of the standard deviations of PSO, BAT, and BATDQL.

From instances ten to sixteen, there are slight variations in the solutions obtained by each metaheuristic, although the value of the best solution remains identical in most instances. As for the worst value generated, variations begin to develop, causing variations to appear in the average, standard deviation, median, and interquartile range. In metaheuristics improved with Deep Q-Learning, specifically PSODQL, BATDQL, and OPADQL, it is verified that the standard deviation is lower compared to their corresponding native metaheuristics. This exciting finding demonstrates that the solution values are very similar to the average; in other words, there is little variability among the solution results, suggesting that the results are consistent and stable. Moreover, experiments with Deep Q-Learning metaheuristics indicate that the experiments are reliable and that random errors have a minimal impact on the outcomes.

Subsequently, in instance seventeen, a great variety is observed in the solutions generated, with PSODQL providing the best solution and OPADQL in second place, maintaining the previous finding with respect to the standard deviation.

For instance, from eighteen to twenty, there is a wide variety of solutions, highlighting PSODQL, BATDQL, and OPADQL. It is interesting to verify that BAT, GWO, and OPA, both native and improved, generate the exact value of the best solution. However, the standard deviation in the improved metaheuristics is lower than that obtained in the native metaheuristics, which reaffirms the consistency and stability of the results.

From instance twenty-one to instance thirty-two, the PSODQL, BATDQL, and OPADQL metaheuristics generate better solution values concerning their corresponding native metaheuristics, and regarding their corresponding standard deviations, they are lower concerning the native metaheuristics; PSODQL’s performance stands out as it produces the best solution values.

In instances thirty-three and thirty-four, the performance of the metaheuristics PSODQL, BATDQL, and OPADQL is maintained, highlighting the excellent performance of BATDQL in instance thirty-three and OPADQL in instance thirty-four.

Concluding with instances thirty-five to forty, we can observe that PSODQL, BATDQL, and OPADQL continue to obtain the best solution values; the standard deviations maintain a small value compared to their native counterparts, and PSODQL, which generated the best solution value, is highlighted.

In the application of metaheuristics with Deep Q-Learning, specifically PSODQL, BATDQL, and OPADQL, in addition to generating better solution values, observing a low standard deviation is beneficial as it indicates that the generated solutions are efficiently clustered around optimal values, thus reflecting the high precision and consistency of the results. This pattern suggests the algorithms’ notable effectiveness in identifying optimal or near-optimal solutions, with minimal variation across multiple executions, a crucial aspect for effectively resolving complex problems. Furthermore, a reduced interquartile range reaffirms the concentration of solutions around the median, decreasing data dispersion and refining the search towards regions of the solution space with high potential, which improves precision in reaching efficient solutions.

To present the results graphically, we faced the challenge of analyzing and comparing samples generated from non-parametric underlying processes, that is, processes whose data do not assume a normal distribution. Given this, it became essential to use a visualization tool such as the violin diagram, which adequately handles the non-parametric nature of the data and provides a clear and detailed view of their corresponding distributions. Visualizing these graphs allows us to consolidate the previously analyzed results, corresponding to the evaluation metric and, later in this section, the Wilcoxon–Mann–Whitney test.

Figure 5, Figure 6, Figure 7 and Figure 8 enrich our comprehension of the effectiveness of biomimetic algorithms (left side) and their enhanced version (right side). These graphical illustrations reveal the data’s distribution; highlighting that the learning component provides a real improvement for each optimization algorithm. The violin diagram is an analytical tool that combines box plots and kernel density diagrams to compare data distribution between two samples; it was used to visualize the results. It shows summarized statistics, such as medians and quartiles, and the data density along its range. It helps identify and analyze significant differences between two samples, offering insights into patterns and the data structure [78]. This way, we can appreciate that in instances fifteen and sixteen, the standard deviation is small in the metaheuristics with DQL compared to native metaheuristics, especially PSODQL, BATDQL, and OPADQL. Furthermore, the median in PSODQL in instance fifteen is much lower than in native PSO. For instances seventeen to twenty, in addition to noting the minor standard deviation in the metaheuristics with Q-Learning, the medians for PSODQL and OPADQL are significantly lower than their native counterparts. From twenty to twenty-six, the previous results for the metaheuristics with DQL are maintained, and the distributions and medians for PSODQL and OPADQL move to lower values. For instances twenty-seven and twenty-eight, the standard deviation is small in the metaheuristics with DQL compared to the native metaheuristics. For instance, we can verify that the distribution and the median in PSODQL reach lower values in twenty-nine. For instances thirty and thirty-one, the distributions and medians in PSODQL, BATDQL, and OPADQL reach lower values. For instance thirty-two, both PSODQL and OPADQL distributions and medians reach lower values, and from thirty-three to forty, we can verify that in most cases, PSODQL, BATDQL, and OPA’s medians tend to lower values. From the above, we can confirm that the visualizations of the solutions for the instances allow us to reaffirm the findings and results of the substantial improvement of the metaheuristics with DQL compared to the native metaheuristics, highlighting PSODQL as the one that generates the best solutions throughout the experimentation phase.

It is worth mentioning that the visualization of the solutions from instances one to fourteen is impossible to graph, since they mainly generate the same statistical values.

In the context of this research, we conducted the Wilcoxon–Mann–Whitney test, a non-parametric statistical test used to compare two independent samples [79]. It was used to determine if there were significant differences in two groups of samples that may not have the same distribution, which were generated from native metaheuristics and DQL. The significance level was previously set at *p* = 0.05 to conduct the test.

The results are detailed in Table 8, describing the following findings. It is verified that from instances fifteen and sixteen, there are significant differences between the samples generated by PSODQL and native PSO, concluding that there is an improvement in the results obtained by PSODQL. For BAT and BATDQL, there are no significant differences between the samples; the same is true for GWO and GWODQL. However, for OPA and OPADQL, there is a substantial difference between the samples. PSODQL shows a more remarkable improvement, as it has a more significant difference than OPADQL since the obtained *p*-value is lower, as verified in the table. In instances seventeen, for the samples generated by PSO, there is a significant difference between the samples, resulting in a better performance of PSODQL; the same happens with BAT, resulting in a better BATDQL; for native GWO, it is better than GWODQL, and for the samples generated by OPA, there is a significant difference, resulting in a better OPADQL. In the eighteenth and nineteenth instances, it is confirmed that PSODQL is better than PSO. For BAT and BATDQL, there are no significant differences between the samples, just as for GWO and GWODQL. Moreover, OPADQL is better for OPA, as there are substantial differences between the samples. In both cases, PSODQL is better since it has the lowest *p* value. In instance twenty, PSODQL and OPADQL show significant differences between their samples; however, OPADQL is better since it has the lowest *p* value. In instance twenty-one, given the obtained results, PSODQL is better than native PSO, and the same applies to BAT; for GWO, native GWO is better, and for OPADQL, there are no significant differences between the samples. For this instance, PSODQL is better since it has the lowest *p* value.

For instances from twenty-three to twenty-eight, significant differences are verified between the samples generated by PSO, BAT, and OPA, the result being that the samples generated by DQL are better. For GWO, there are cases of significant differences between their samples. For the twenty-ninth instance, significant differences exist for the samples generated by PSO and OPA, resulting in better PSODQL and OPADQL. For instances thirty to thirty-two, PSODQL, BATDQL, and OPADQL are better. For the thirty-third instances, PSODQL and OPADQL turned out to be better. For instances thirty-four to thirty-six, PSODQL, BATDQL, and OPADQL are better. For the thirty-seventh instance, PSODQL and OPADQL are better. PSODQL, BATDQL, and OPADQL are the best for the thirty-eighth cases. For instances thirty-nine and forty, PSODQL and OPADQL are the best.

The central objective of this study was to evaluate the impact of integrating the Deep Q-Learning technique into traditional metaheuristics to improve their effectiveness in optimization tasks. The results demonstrate that the Deep Q-Learning-enhanced versions, specifically PSODQL, BATDQL, and OPADQL, exhibited superior performance compared to their native counterparts. Notably, PSODQL stood out significantly, outperforming native PSO in one hundred percent of the cases during the experimental phase. These findings highlight the potential of reinforcement learning through Deep Q-Learning as an effective strategy to enhance the performance of metaheuristics in optimization problems.

## 7. Conclusions

The presented research tackles the challenge of enhancing the efficiency of Cybersecurity Operations Centers through the integration of biomimetic algorithms and Deep Q-Learning, a reinforcement learning technique. This approach is proposed to improve the deployment of sensors across network infrastructures, balancing security imperatives against deployment costs. The research is grounded in the premise that the dynamic nature of cyber threats necessitates adaptive and efficient solutions for cybersecurity management.

The study demonstrated that incorporating DQL into biomimetic algorithms significantly improves the effectiveness of these algorithms, enabling optimal resource allocation and efficient intrusion detection. Experimental results validated the hypothesis that combining biomimetic optimization techniques with deep reinforcement learning leads to superior solutions compared to conventional strategies.

A comparative analysis between native biomimetic algorithms and those enhanced with DQL revealed a notable improvement in the accuracy and consistency of the solutions obtained. This enhancement is attributed to the ability of DQL to dynamically adapt and fine-tune the algorithms’ parameters, focusing the search towards the most promising regions of the solution space. Moreover, the implementation of replay memory and the mini-batch strategy in DQL contributed to learning efficiency and training stability.

The study underscores the importance of integrating machine learning techniques with optimization algorithms to address complex problems in cybersecurity. The adaptability and improved performance of biomimetic algorithms enhanced with DQL offer a promising approach to efficient Cyber SOC management, highlighting the potential of these advanced techniques in the cybersecurity domain.

Future works could pivot towards creating adaptive defense mechanisms by integrating biomimetic algorithms with Deep Q-Learning, focusing on real-time threat responses and evolutionary security frameworks. This would entail embedding ethical AI principles to ensure that these advanced systems operate without bias and respect privacy. Additionally, exploring federated learning for collaborative defense across Cyber SOCs could revolutionize how threat intelligence is shared, fostering a unified global response to cyber threats without compromising sensitive data. These directions promise to significantly elevate the cybersecurity landscape, making it more resilient, ethical, and collaborative.

## Figures and Tables

**Figure 1 biomimetics-09-00307-f001:**
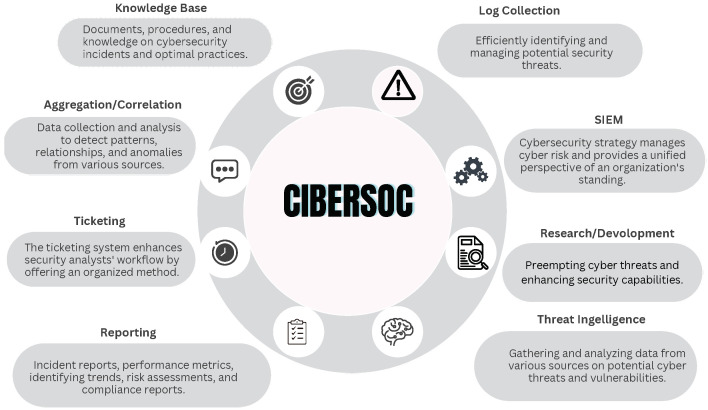
Cyber SOC functional characteristics.

**Figure 2 biomimetics-09-00307-f002:**
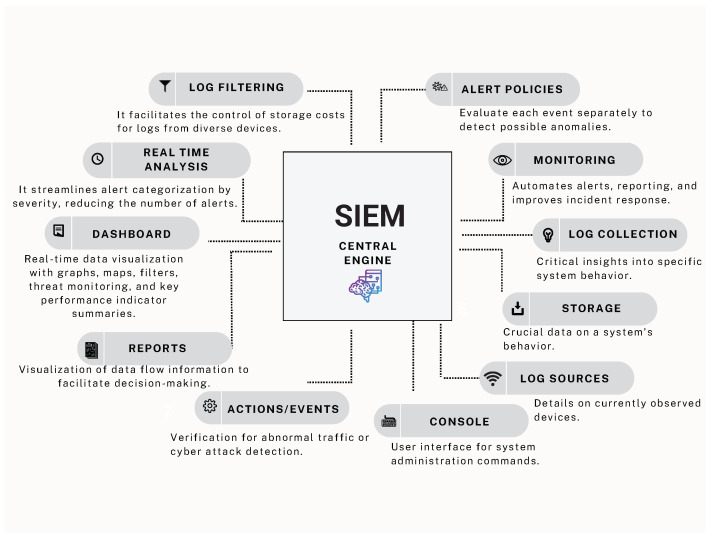
SIEM functional characteristics.

**Figure 3 biomimetics-09-00307-f003:**
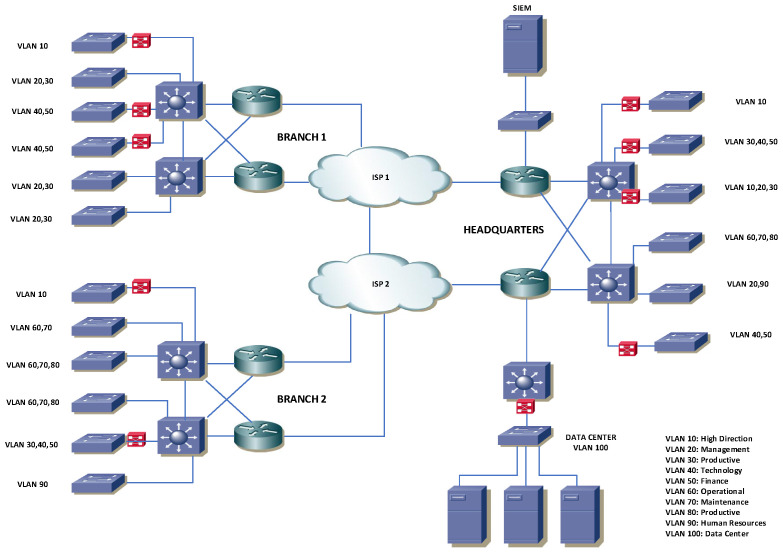
Network topology. Instance-zero solution.

**Figure 4 biomimetics-09-00307-f004:**
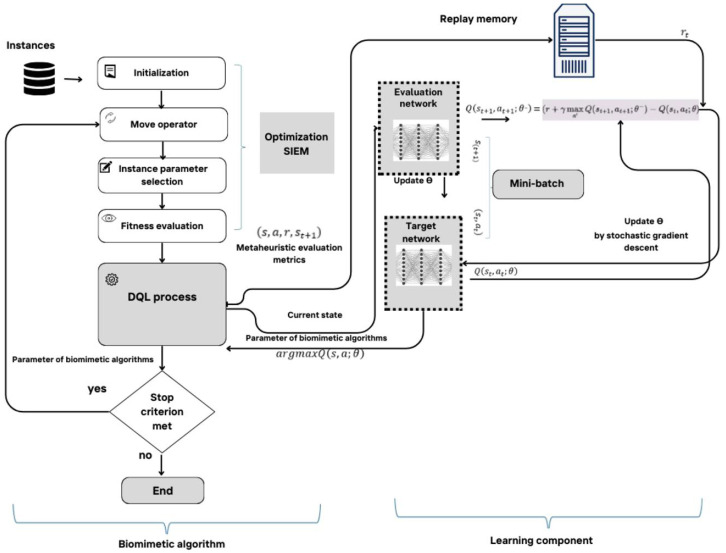
Solution developed using four metaheuristics with Deep Q-Learning.

**Figure 5 biomimetics-09-00307-f005:**
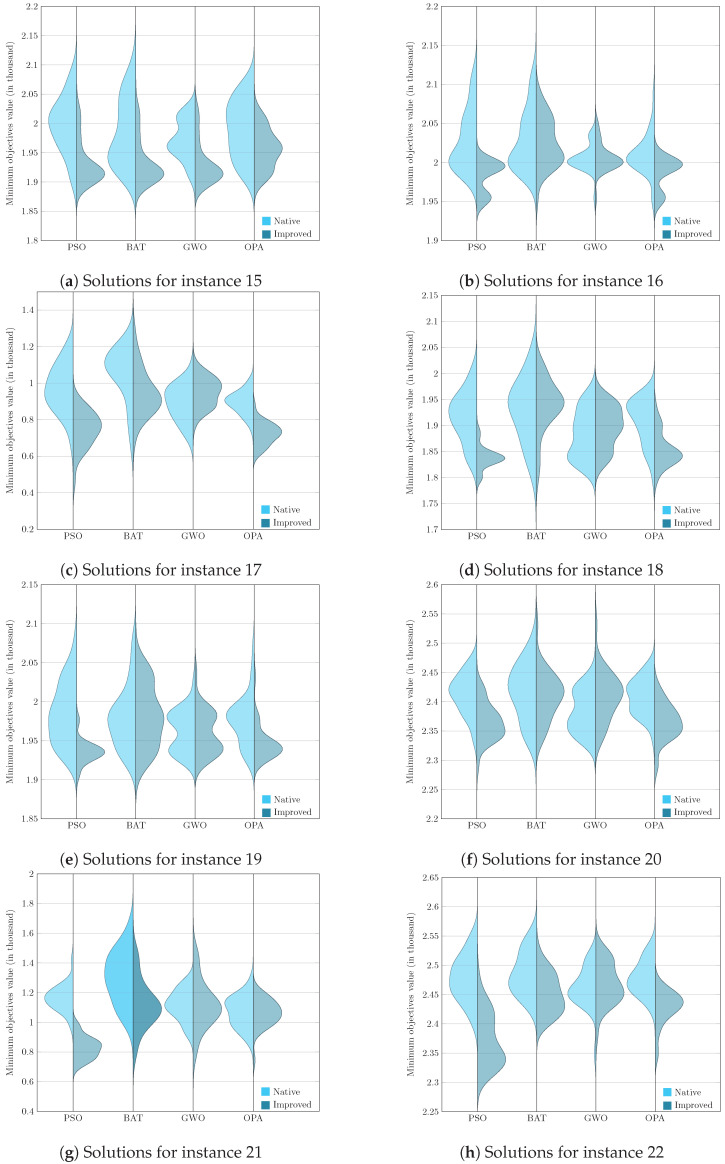
Computational result distributions between improved biomimetic algorithms against their native versions. Hardest instances from 15 to 22.

**Figure 6 biomimetics-09-00307-f006:**
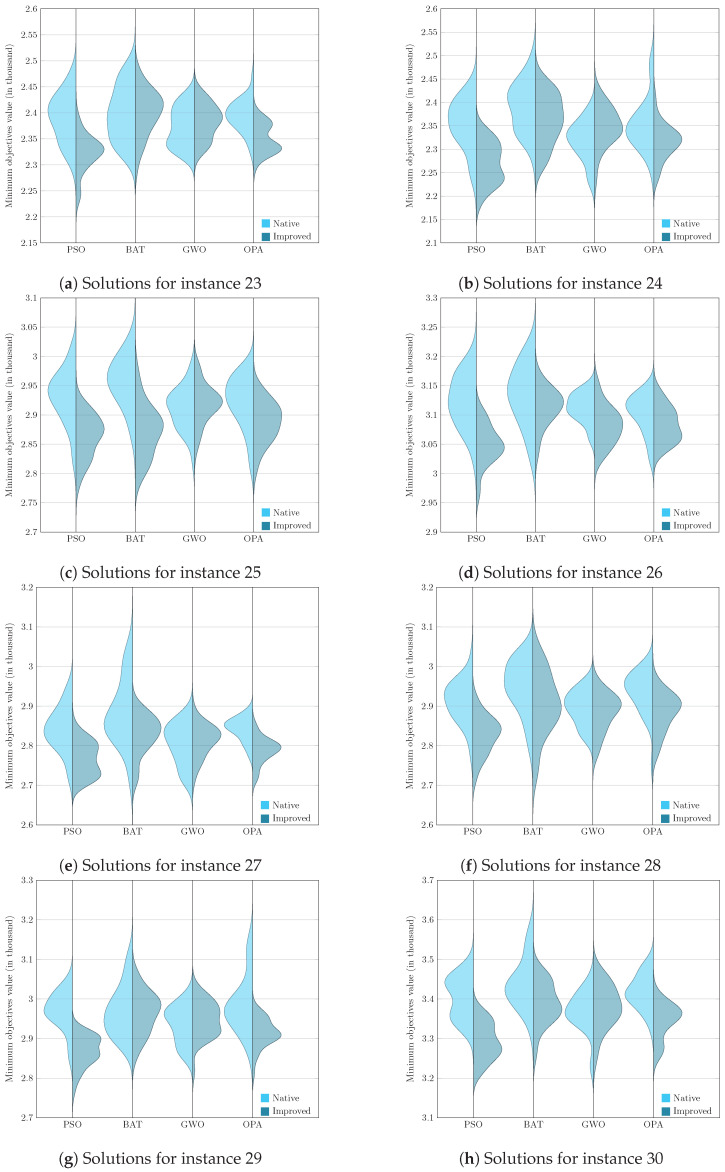
Computational result distributions between improved biomimetic algorithms against their native versions. Hardest instances from 23 to 30.

**Figure 7 biomimetics-09-00307-f007:**
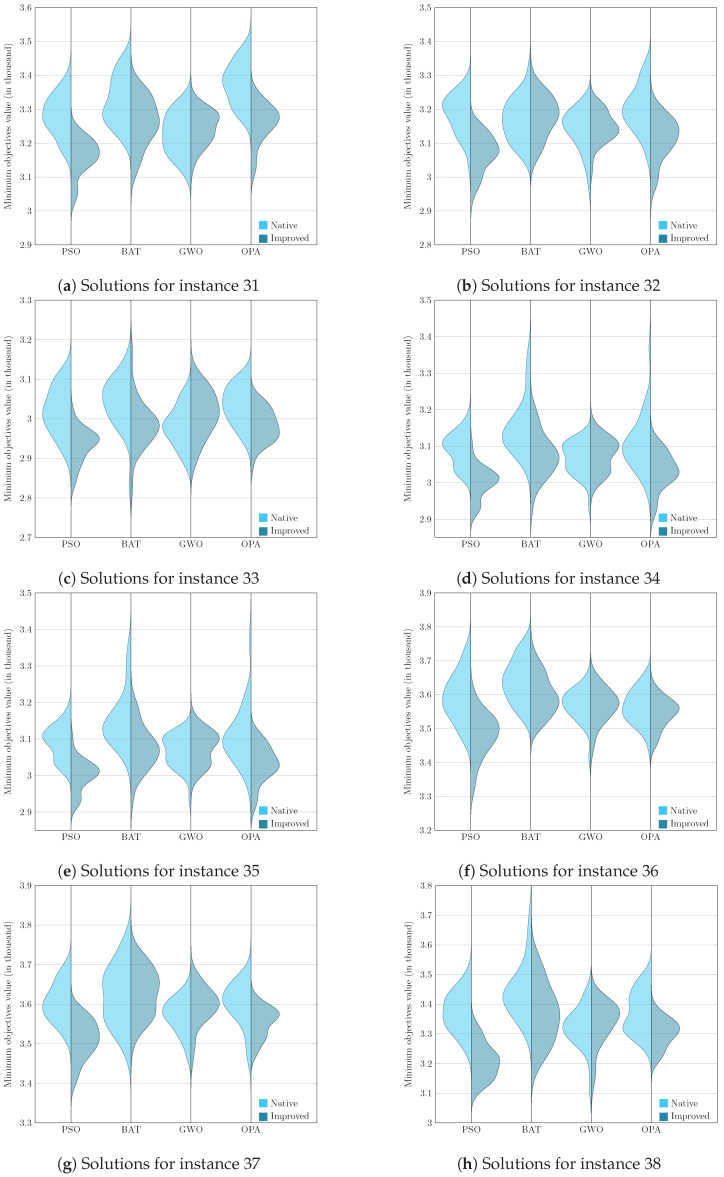
Computational result distributions between improved biomimetic algorithms against their native versions. Hardest instances from 31 to 38.

**Figure 8 biomimetics-09-00307-f008:**
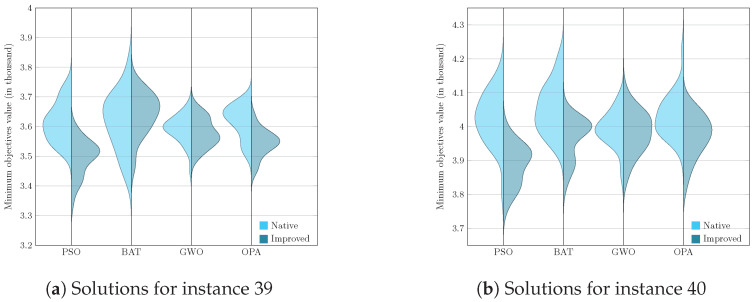
Computational result distributions between improved biomimetic algorithms against their native versions. Hardest instances from 39 to 40.

**Table 1 biomimetics-09-00307-t001:** Specification of the operational parameters for the forty instances.

Instance	Number of VLANs	Type of Sensors	Uptime	Range of Direct Costs	Qualitative Profit Range	Range of Indirect Costs	Performance of Subnets
1	10	2	90%	[100–150]	[1–20]	[1–7]	[0.39–0.80]
2	10	2	90%	[100–150]	[5–20]	[1–7]	[0.10–0.80]
3	10	2	90%	[100–150]	[1–20]	[1–7]	[0.02–0.80]
4	10	2	90%	[100–150]	[1–20]	[1–5]	[0.11–0.80]
5	15	2	90%	[100–150]	[1–20]	[1–7]	[0.14–0.85]
6	15	2	90%	[100–150]	[1–20]	[1–7]	[0.01–0.94]
7	15	2	90%	[100–150]	[1–20]	[1–7]	[0.01–0.94]
8	15	2	90%	[100–150]	[1–20]	[1–7]	[0.07–0.96]
9	15	2	90%	[100–150]	[1–20]	[3–7]	[0.07–0.96]
10	20	2	90%	[100–150]	[1–20]	[1–7]	[0.04–0.61]
11	20	2	90%	[100–150]	[1–20]	[1–7]	[0.07–0.56]
12	20	2	90%	[100–150]	[1–20]	[1–7]	[0.10–0.91]
13	20	2	90%	[100–150]	[1–20]	[1–7]	[0.01–0.99]
14	20	2	90%	[100–150]	[1–20]	[1–7]	[0.05–0.88]
15	25	2	90%	[100–150]	[1–20]	[1–7]	[0.07–0.96]
16	25	2	90%	[100–150]	[1–20]	[1–7]	[0.07–0.96]
17	25	2	90%	[100–150]	[1–20]	[1–7]	[0.07–0.89]
18	25	2	90%	[100–150]	[1–20]	[1–5]	[0.08–0.97]
19	25	2	90%	[100–150]	[1–20]	[1–7]	[0.06–0.99]
20	30	2	90%	[100–150]	[10–20]	[1–7]	[0.50–0.89]
21	30	2	90%	[100–150]	[1–20]	[1–7]	[0.22–0.89]
22	30	2	90%	[100–150]	[1–20]	[1–7]	[0.07–0.96]
23	30	2	90%	[100–150]	[1–20]	[1–7]	[0.08–0.97]
24	30	2	90%	[100–150]	[1–20]	[1–7]	[0.05–0.98]
25	35	2	90%	[100–150]	[1–20]	[1–7]	[0.10–0.96]
26	35	2	90%	[100–150]	[1–20]	[1–7]	[0.07–0.94]
27	35	2	90%	[100–150]	[1–20]	[1–7]	[0.07–0.94]
28	35	2	90%	[100–150]	[1–20]	[1–7]	[0.03–0.98]
29	35	2	90%	[100–150]	[1–20]	[1–7]	[0.08–0.98]
30	40	2	90%	[100–150]	[1–20]	[1–7]	[0.06–0.98]
31	40	2	90%	[100–150]	[1–20]	[1–7]	[0.05–0.98]
32	40	2	90%	[100–150]	[1–20]	[1–7]	[0.04–0.97]
33	40	2	90%	[100–150]	[1–20]	[1–7]	[0.16–0.93]
34	40	2	90%	[100–150]	[1–20]	[1–7]	[0.09–0.95]
35	45	2	90%	[100–150]	[1–20]	[1–7]	[0.01–0.95]
36	45	2	90%	[100–150]	[1–20]	[1–7]	[0.07–0.97]
37	45	2	90%	[100–150]	[1–20]	[1–7]	[0.01–0.95]
38	45	2	90%	[100–150]	[1–20]	[1–7]	[0.03–0.97]
39	45	2	90%	[100–150]	[1–20]	[1–7]	[0.07–0.96]
40	50	2	90%	[100–150]	[1–20]	[1–7]	[0.02–0.84]

**Table 2 biomimetics-09-00307-t002:** Experimental parameters and platform details.

Parameter	Value
Particle Swarm Optimization	
Inertia weight (*w*)	0.6
Cognitive acceleration ($c_{1}$)	0.6
Social acceleration ($c_{2}$)	0.6
Number of particles (ps)	10
Maximum iterations (*T*)	100
Bat Algorithm	
Larger search space jumps for broader exploration ($f_{min}$)	0.75
Finer adjustments for more detailed exploitation ($f_{max}$)	1.25
Modulating the decay rate of loudness over time (α)	0.9
Modulating the pulse rate’s decay over time (γ)	0.9
Adding randomness to the bat’s movement. (ϵ)	0.9
Number of virtual bats (ps)	10
Maximum iterations (*T*)	100
Gray Wolf Optimization	
Number of wolves (ps)	10
Maximum iterations (*T*)	100
Orca Predator Algorithm	
Explorative and exploitative behaviors (*p*)	0.5
Influence of leading orcas on the group’s movement (*q*)	0.75
Attraction force (*F*)	2
Number of orcas (ps)	10
Maximum iterations (*T*)	100
Deep Q-Learning	
Action size	40
Neurons per layer	20
Activation	ReLU (layers), Linear (final layer)
Loss function	Huber
Optimizer	RMSprop with a learning rate of 0.001
Epsilon-greedy	Starts at 1.0, decays to 0.01
Network update	Every 50 training steps
Platform details	
Operating system	macOS 14.2.1 Darwin Kernel v23
Programming language	Python 3.10
Hardware specifications	Ultra M2 chip, 64 GB RAM

**Table 3 biomimetics-09-00307-t003:** Comparison between improved biomimetic algorithms against their native versions. Instances from 1 to 8.

Instances	Metrics	Native Algorithms				Improved Algorithms			
		PSO	BAT	GWO	OPA	PSODQL	BATDQL	GWODQL	OPADQL
1	Best	950	950	950	950	950	950	950	950
	Worst	950	950	950	950	950	950	950	950
	Mean	950	950	950	950	950	950	950	950
	Std	0	0	0	0	0	0	0	0
	Median	950	950	950	950	950	950	950	950
	Iqr	0	0	0	0	0	0	0	0
2	Best	773	773	773	773	773	773	773	773
	Worst	773	773	773	773	773	773	773	773
	Mean	773	773	773	773	773	773	773	773
	Std	0	0	0	0	0	0	0	0
	Median	773	773	773	773	773	773	773	773
	Iqr	0	0	0	0	0	0	0	0
3	Best	822	822	822	822	822	822	822	822
	Worst	822	822	822	822	822	822	822	822
	Mean	822	822	822	822	822	822	822	822
	Std	0	0	0	0	0	0	0	0
	Median	822	822	822	822	822	822	822	822
	Iqr	0	0	0	0	0	0	0	0
4	Best	872	872	872	872	872	872	872	872
	Worst	872	872	872	872	872	872	872	872
	Mean	872	872	872	872	872	872	872	872
	Std	0	0	0	0	0	0	0	0
	Median	872	872	872	872	872	872	872	872
	Iqr	0	0	0	0	0	0	0	0
>5	Best	1215	1215	1215	1215	1215	1215	1215	1215
	Worst	1215	1253	1215	1215	1215	1215	1215	1215
	Mean	1215	1244.2	1215	1215	1215	1215	1215	1215
	Std	0	14.1	0	0	0	0	0	0
	Median	1215	0	1215	1215	1215	1215	1215	1215
	Iqr	0	25	0	0	0	0	0	0
6	Best	1235	1235	1235	1235	1235	1235	1235	1235
	Worst	1318	1397	1235	1235	1235	1318	1235	1235
	Mean	1238.50	1282.80	1235	1235	1235	1238.87	1235	1235
	Std	15.27	63.02	0	0	0	15.32	0	0
	Median	1235	1235	1235	1235	1235	1235	1235	1235
	Iqr	0	93	0	0	0	0	0	0
7	Best	1287	1287	1287	1287	1287	1287	1287	1287
	Worst	1287	1287	1287	1287	1287	1287	1287	1287
	Mean	1287	1287	1287	1287	1287	1287	1287	1287
	Std	0	0	0	0	0	0	0	0
	Median	1287	1287	1287	1287	1287	1287	1287	1287
	Iqr	0	0	0	0	0	0	0	0
8	Best	1269	1269	1269	1269	1269	1269	1269	1269
	Worst	1284	1303	1269	1269	1269	1269	1269	1269
	Mean	1270.30	1278.33	1269	1269	1269	1269	1269	1269
	Std	3.99	14.47	0	0	0	0	0	0
	Median	1269	1269	1269	1269	1269	1269	1269	1269
	Iqr	0	19.75	0	0	0	0	0	0

**Table 4 biomimetics-09-00307-t004:** Comparison between improved biomimetic algorithms against their native versions. Instances from 9 to 16.

Instances	Metrics	Native Algorithms				Improved Algorithms			
		PSO	BAT	GWO	OPA	PSODQL	BATDQL	GWODQL	OPADQL
9	Best	1303	1303	1303	1303	1303	1303	1303	1303
	Worst	1305	1303	1303	1303	1303	1303	1303	1303
	Mean	1303.07	1303	1303	1303	1303	1303	1303	1303
	Std	0.37	0	0	0	0	0	0	0
	Median	1303	1303	1303	1303	1303	1303	1303	1303
	Iqr	0	0	0	0	0	0	0	0
10	Best	1536	1536	1536	1536	1536	1536	1536	1535
	Worst	1636	1737	1592	1596	1547	1679	1596	1547
	Mean	1560.73	1584.53	1543.07	1551.10	1536.37	1569.57	1548.77	1536.70
	Std	27.54	56.93	14.87	21.86	2.01	35.20	19.79	2.81
	Median	1547	1547	1536	1536	1536	1564	1541.50	1536
	Iqr	45	92.75	11	45	0	56	11	0
11	Best	1593	1593	1593	1593	1593	1593	1593	1593
	Worst	1687	1690	1607	1650	1593	1681	1641	1599
	Mean	1606.17	1607.17	1593.87	1597.20	1593	1600.90	1595.40	1593.20
	Std	23.76	30.52	2.91	13.59	0	19.56	8.86	1.10
	Median	1596	1593	1593	1593	1593	1593	1593	1593
	Iqr	14	6	0	0	0	6	0	0
12	Best	1608	1608	1608	1608	1608	1608	1608	1608
	Worst	1689	1703	1642	1658	1642	1689	1642	1615
	Mean	1629	1628.87	1611	1616.67	1611.63	1633.03	1611	1608
	Std	21.73	26.63	6.64	14.60	10.37	25.23	6.64	1.78
	Median	1615	1615	1611	1608	1608	1642	1608	1608
	Iqr	40	40	7	7	0	42	7	0
13	Best	1530	1530	1530	1530	1530	1530	1530	1530
	Worst	1632	1626	1537	1568	1531	1633	1566	1537
	Mean	1547.47	1548.70	1530.93	1537.53	1530.03	1545.53	1535.13	1530.37
	Std	27.21	28.31	2.15	11.36	0.18	27.36	8.18	1.30
	Median	1535	1535	1530	1535	1530	1531	1533	1530
	Iqr	35.25	36	1.00	7	0	29	7	0
14	Best	1449	1449	1449	1449	1449	1449	1449	1449
	Worst	1588	1609	1507	1549	1497	1559	1540	1508
	Mean	1498.43	1495.03	1462.20	1486.43	1452.27	1490.53	1488.57	1452.30
	Std	45.64	46.67	21.51	35.07	9.25	39.62	33.31	11.07
	Median	1497	1478	1449	1497	1449	1497	1497	1449
	Iqr	90	82	20	58	0	89	70	0
15	Best	1910	1920	1910	1920	1910	1910	1910	1910
	Worst	2089	2105	2018	2062	2020	2020	2180	2018
	Mean	1997.97	1984.87	1972.10	1989.97	1931.27	1980.83	1985.50	1956.30
	Std	46.72	56.33	32.60	47.75	34.67	32.78	54.72	32.59
	Median	2008	1965	1966	1999	1915	1999	1980	1956
	Iqr	49.25	97	54.25	73.50	21.50	38	54.75	53.75
16	Best	1994	1994	1955	1955	1955	1955	1955	1955
	Worst	2112	2112	2037	2087	2001	2087	2049	2012
	Mean	2028.27	2028.27	2005.23	2009.50	1982.03	2025.30	2006.13	1986.60
	Std	37.28	37.28	16.32	25.45	19.50	30.67	18.28	19.73
	Median	2005	2005	2001	2005	1994	2012	2001	1996
	Iqr	56.50	57	12.25	20	41	51	12	42

**Table 5 biomimetics-09-00307-t005:** Comparison between improved biomimetic algorithms against their native versions. Instances from 17 to 24.

Instances	Metrics	Native Algorithms				Improved Algorithms			
		PSO	BAT	GWO	OPA	PSODQL	BATDQL	GWODQL	OPADQL
17	Best	675	651	683	749	461	698	781	609
	Worst	1203	1256	1089	1011	911	1284	1098	878
	Mean	971.83	1054.53	890.77	893.57	744.37	948.27	950.03	723.30
	Std	126.54	144.85	97.83	61.44	100.04	125.35	77.97	60.57
	Median	969	1075.50	897.50	901.50	759	929	958	724
	Iqr	175.75	150	159.50	74.75	170.50	182.25	139.50	98
18	Best	1832	1801	1832	1801	1801	1801	1832	1801
	Worst	2005	2044	1950	1958	1889	2009	1949	1936
	Mean	1915.90	1930.03	1881.80	1909.27	1837.60	1938.90	1897.13	1853.30
	Std	47.28	60.53	44.64	42.16	20.79	50.13	38.80	31.52
	Median	1918.50	1937	1887	1927	1835	1945	1898.50	1849
	Iqr	51.50	77.25	86	60.25	8.75	26	86.50	25.25
19	Best	1935	1930	1930	1930	1905	1930	1930	1930
	Worst	2074	2075	2024	2069	1978	2042	2036	2032
	Mean	1984.13	1979.43	1962.27	1976.07	1936.13	1981.30	1960.23	1947.33
	Std	38.19	38.90	25.70	31.51	14.55	37.87	26.71	21.64
	Median	1981	1978	1962.27	1978.50	1935	1979.50	1947	1942
	Iqr	80.50	51.50	49	42	12	82	42.75	12
20	Best	2337	2331	2334	2339	2293	2337	2334	2293
	Worst	2470	2507	2450	2459	2426	2532	2437	2428
	Mean	2413.33	2416.10	2383.20	2408.90	2364.77	2410.20	2390.63	2367.38
	Std	33.92	47.48	35.02	30.34	30.60	41.58	27.87	28.53
	Median	2419.50	2423	2374	2413.50	2366	2416	2384	2365
	Iqr	50.50	76.75	68.25	46.25	36.75	55.25	41	38.25
21	Best	973	978	915	868	702	805	896	745
	Worst	1419	1648	1277	1289	980	1455	1321	1196
	Mean	1161.90	1303.63	1114.03	1070.77	829.53	1119.20	1196.10	1061.37
	Std	86.07	173.08	100	103.82	73.28	156.64	91.17	93.70
	Median	1161	1291	1108.50	1081.50	829	1102	1207.50	1076
	Iqr	101	312.75	124	172.25	110	154	105	119
22	Best	2400	2423	2349	2423	2323	2400	2384	2349
	Worst	2596	2557	2520	2538	2473	2524	2529	2467
	Mean	2484.10	2486.43	2462.13	2478	2372.20	2450.77	2468.93	2427.70
	Std	46.43	36.82	34.09	29.08	43.39	33.11	34.70	28.82
	Median	2477	2480.50	2456.50	2477	2357.50	2447	2464.50	2431.50
	Iqr	58	69.75	43.50	41.25	84	55	63.25	24.50
23	Best	2295	2319	2323	2323	2244	2295	2326	2297
	Worst	2478	2489	2430	2469	2390	2474	2436	2399
	Mean	2390.50	2398.03	2372.03	2393.80	2322	2400.83	2383.50	2349.60
	Std	45.84	51.33	36.54	30.29	34.66	42.41	32.66	27.03
	Median	2391	2390.50	2373.50	2394	2328	2410.50	2390.50	2338
	Iqr	78	93.75	70	39	37.75	55.25	57.25	48.25
24	Best	2232	2279	2228	2248	2193	2275	2238	2238
	Worst	2439	2502	2386	2491	2337	2434	2423	2411
	Mean	2354.50	2391.40	2317.87	2352.63	2269.77	2363.13	2350	2315.57
	Std	52.07	52.94	42	54.69	42.39	51.75	41.08	36.85
	Median	2363.50	2397	2327	2342	2279	2368	2342.50	2320
	Iqr	62.25	99	57.75	52	76	92	50	39.50

**Table 6 biomimetics-09-00307-t006:** Comparison between improved biomimetic algorithms against their native versions. Instances from 25 to 32.

Instances	Metrics	Native Algorithms				Improved Algorithms			
		PSO	BAT	GWO	OPA	PSODQL	BATDQL	GWODQL	OPADQL
25	Best	2826	2872	2821	2817	2782	2791	2842	2805
	Worst	3008	3176	2984	2975	2922	2976	2980	2946
	Mean	2931.90	2964.93	2910.47	2921.67	2863.03	2872.33	2919.37	2887.87
	Std	44.78	55.33	36.04	43.44	35.78	43.97	32.81	34.57
	Median	2936.50	2959	2915.50	2931	2870	2874	2924	2887.50
	Iqr	51.75	57	52.50	64.50	63.25	67	31.75	46.50
26	Best	3022	3018	3056	3022	2968	3016	3024	3037
	Worst	3208	3220	3152	3144	3106	3172	3155	3141
	Mean	3123.80	3133.20	3112.17	3101.27	3048.20	3114.20	3081.70	3083.27
	Std	44.70	51.76	24.72	33.46	34.13	31.64	32.36	29.32
	Median	3121	3140	3112.50	3109	3048.50	3119.50	3083	3081
	Iqr	72.50	78.75	35	49.75	37.75	36	44	45.50
27	Best	2724	2721	2716	2775	2711	2714	2714	2717
	Worst	2948	3052	2884	2886	2848	2891	2889	2851
	Mean	2840.90	2882.73	2803.47	2841.56	2765.87	2827.27	2813.27	2787.63
	Std	55.32	81.30	51.33	29.33	41.07	48.77	39.28	32.25
	Median	2842	2858.50	2811.50	2851	2767.50	2840.50	2821.50	2794
	Iqr	64.25	96	75.75	33.25	78.75	64	61.25	32.25
28	Best	2776	2790	2817	2752	2714	2727	2778	2717
	Worst	3024	3037	2960	3005	2915	3045	2963	2911
	Mean	2911.20	2949.43	2894.97	2935.37	2825.77	2905.10	2883.17	2827.50
	Std	52.22	65.12	42.59	53.77	48	79.31	48.04	50.01
	Median	2919.50	2954.50	2903.50	2945.50	2831	2894.50	2895.50	2832
	Iqr	73.75	90.75	63.75	66.50	67.25	114.50	73.50	59
29	Best	2850	2870	2859	2859	2770	2859	2820	2813
	Worst	3051	3108	3007	3151	2913	3060	3026	2973
	Mean	2970	2968.83	2935.23	2984.43	2868.07	2965.23	2947.90	2914.27
	Std	47.16	63	46.58	70.05	46.72	51.64	44.59	37.74
	Median	2966	2957.50	2953	2971.50	2862	2971.50	2956	2910
	Iqr	49.75	92.50	65.75	97	53.50	83.50	75.75	43.50
30	Best	3314	3310	3231	3318	3218	3257	3247	3258
	Worst	3487	3579	3457	3503	3367	3471	3470	3409
	Mean	3403.63	3438.47	3368.17	3417.77	3293.03	3388.17	3380	3344.23
	Std	51.38	63.38	50.04	43.61	45.12	52.42	54.39	43.35
	Median	3417	3435	3368	3413	3285	34.13	3380	3356.50
	Iqr	96	105	58.75	58	74.25	81	70.25	62
31	Best	3179	3224	3109	3274	3044	3119	3133	3116
	Worst	3399	3447	3328	3475	3232	3357	3319	3341
	Mean	3281.73	3325.80	3226.70	3375.37	3160.53	3262.17	3246.50	3258.23
	Std	58.10	69.52	60.41	59.75	51.73	68.12	47.39	57.50
	Median	3276	3307.50	3230	3380.50	3162	3261	3250.50	3272
	Iqr	73.50	105	103.50	102	61.50	96.50	71.25	64
32	Best	3034	3062	2973	3063	2946	3057	3026	2974
	Worst	3276	3317	3220	3329	3145	3283	3218	3206
	Mean	3185.63	3168.50	3138.90	3200.73	3071.10	3177.27	3148.17	3113.77
	Std	57.72	62.22	59.77	65.93	53.06	59.78	44.15	61.49
	Median	3195.50	3167.50	3152	3200	3074.50	3182.50	3145	3129.50
	Iqr	88.50	109	82.25	85	65	102	62	71.50

**Table 7 biomimetics-09-00307-t007:** Comparison between improved biomimetic algorithms against their native versions. Instances from 33 to 40.

Instances	Metrics	Native Algorithms				Improved Algorithms			
		PSO	BAT	GWO	OPA	PSODQL	BATDQL	GWODQL	OPADQL
33	Best	2904	2897	2890	2938	2838	2830	2896	2838
	Worst	3119	3188	3076	3105	3017	3167	3112	3017
	Mean	3019.31	3020.73	2980.83	3034.27	2932.40	2984.07	3014.40	2932.40
	Std	58.95	74.00	49.47	47.73	41.58	58.73	57.03	41.58
	Median	3023	3008.50	2983.50	3034.50	2944	2985	3016	2944
	Iqr	91.50	123.25	59	73	56	55	82.75	56
34	Best	3019	3035	2939	2979	2929	2938	2989	2918
	Worst	3183	3357	3125	3370	3102	3203	3157	3108
	Mean	3091.60	3153.33	3065.60	3107.80	2999.93	3074.67	3076.63	3032
	Std	44.17	74.89	43.87	74.72	41.22	58.83	44.21	50.01
	Median	3099	3134	3070.50	3099	3008.50	3067.50	3087	3028.50
	Iqr	78	144	73.50	56	38.25	67.25	76.50	56.25
35	Best	3209	3317	3219	3217	3110	3260	3251	3160
	Worst	3496	3647	3497	3463	3328	3538	3451	3437
	Mean	3394.87	3470.57	3343.53	3359.83	3236.83	3386.90	3350.43	3301.43
	Std	77.42	91.80	57.90	69.65	60.82	85.71	52.28	67.38
	Median	3407	3474.50	3338.50	3387.50	3247.50	3376	3354	3309.50
	Iqr	105.25	159	73.75	119.75	118	160.75	83.75	92.75
36	Best	3475	3537	3416	3448	3331	3509	3438	3456
	Worst	3725	3743	3663	3654	3602	3728	3649	3625
	Mean	3592	3640.40	3570.20	3564.37	3480.73	3597.03	3567.60	3545.17
	Std	68.66	61.44	47.41	49.92	64.01	57.92	49.75	43.33
	Median	3583	3643	3572.50	3559.50	3477.50	3586	3573.50	3556.50
	Iqr	92	85.50	55	73.50	70.50	97.25	72	50.50
37	Best	3516	3478	3471	3463	3408	3509	3462	3473
	Worst	3694	3752	3653	3679	3581	3722	3675	3615
	Mean	3604.63	3621.40	3569.67	3600.67	3511.90	3630.13	3595.97	3553.27
	Std	48.01	73.97	43.76	53.01	47.45	58.41	47.43	38.00
	Median	3605	3624.50	3574.50	3612	3519	3640.50	3601	3566
	Iqr	82.75	111.50	62.25	63.25	78	98.25	56.25	59.75
38	Best	3248	3307	3109	3256	3114	3211	3134	3211
	Worst	3504	3703	3445	3491	3331	3559	3416	3392
	Mean	3372.87	3443.80	3321.23	3371.87	3208.33	3367.83	3334.70	3302.03
	Std	65.74	90.01	68.03	68.50	57.96	96.98	71.81	47.40
	Median	3370	3429	3331	3355.50	3210	3365	3351.50	3310
	Iqr	99	96	73	105.50	81.50	146.25	100.25	76
39	Best	3519	3434	3452	3502	3346	3472	3495	3449
	Worst	3738	3811	3680	3690	3590	3751	3669	3635
	Mean	3613.67	3617.33	3592.20	3617.93	3495.60	3642.90	3579.27	3543.67
	Std	63.44	95.11	45.82	54.35	58.84	69.78	47.05	44.80
	Median	3609	3623	3598	3634	3512.50	3654.50	3575	3546.50
	Iqr	93.25	141	46.75	77.50	85.25	98.50	84.25	60
40	Best	3813	3940	3850	3919	3758	3826	3842	3820
	Worst	4171	4203	4094	4225	3999	4047	4106	4065
	Mean	4025.90	4052.07	3993.17	4031.60	3886.57	3963.47	3982.40	3972.60
	Std	74.31	71.92	52.11	62.17	59.72	59.89	61.02	61.75
	Median	4025.50	4046	3991	4022	3906	4001.50	3976.50	3982
	Iqr	123.75	104.75	62.25	82	96	107	78.75	76

**Table 8 biomimetics-09-00307-t008:** *p*-values obtained from Wilcoxon–Mann–Whitney Test.

Instances	PSO v/s PSODQL	PSODQL v/s PSO	BAT v/s BATDQL	BATDQL v/s BAT	GWO v/s GWODQL	GWODQL v/s GWO	OPA v/s OPADQL	OPADQL v/s OPA
15	–	$1.4\times 10^{-12}$	–	–	–	–	–	$1.5\times 10^{-3}$
16	–	$6.5\times 10^{-15}$	–	–	–	–	–	$1.5\times 10^{-12}$
17	–	$4.2\times 10^{-16}$	–	$7.1\times 10^{-4}$	$1.1\times 10^{-2}$	–	–	$2.4\times 10^{-15}$
18	–	$4.8\times 10^{-16}$	–	–	–	–	–	$1.1\times 10^{-13}$
19	–	$3.7\times 10^{-15}$	–	–	–	–	–	$4.6\times 10^{-12}$
20	–	$1.8\times 10^{-13}$	–	–	–	–	–	$2.6\times 10^{-13}$
21	–	$4.2\times 10^{-15}$	–	$4.5\times 10^{-3}$	$7.1\times 10^{-4}$	–	–	–
22	–	$4.1\times 10^{-17}$	–	$3.5\times 10^{-4}$	–	–	–	$1.8\times 10^{-15}$
23	–	$2.3\times 10^{-14}$	–	–	–	–	–	$7.1\times 10^{-14}$
24	–	$7.6\times 10^{-15}$	–	$3.9\times 10^{-2}$	$2.3\times 10^{-3}$	–	–	$1.1\times 10^{-3}$
25	–	$5.7\times 10^{-15}$	–	$1.2\times 10^{-15}$	–	–	–	$5.1\times 10^{-4}$
26	–	$1.9\times 10^{-15}$	–	$3.6\times 10^{-2}$	–	$1.1\times 10^{-12}$	–	$8.6\times 10^{-3}$
27	–	$4.4\times 10^{-13}$	–	$3.5\times 10^{-3}$	–	–	–	$6.7\times 10^{-15}$
28	–	$7.7\times 10^{-15}$	–	$9.1\times 10^{-3}$	–	–	–	$3.1\times 10^{-16}$
29	–	$2.3\times 10^{-16}$	–	–	–	–	–	$3.1\times 10^{-13}$
30	–	$2.4\times 10^{-16}$	–	$2.7\times 10^{-3}$	–	–	–	$7.6\times 10^{-14}$
31	–	$1.7\times 10^{-16}$	–	$8.4\times 10^{-4}$	–	–	–	$3.2\times 10^{-14}$
32	–	$1.1\times 10^{-15}$	–	$9.4\times 10^{-12}$	–	–	–	$4.6\times 10^{-13}$
33	–	$2.1\times 10^{-14}$	–	$3.1\times 10^{-12}$	$7.7\times 10^{-3}$	–	–	$6.6\times 10^{-12}$
34	–	$1.7\times 10^{-16}$	–	$9.3\times 10^{-13}$	–	–	–	$7.4\times 10^{-13}$
35	–	$4.6\times 10^{-16}$	–	$8.6\times 10^{-4}$	–	–	–	$9.1\times 10^{-4}$
36	–	$8.6\times 10^{-14}$	–	$5.1\times 10^{-3}$	–	–	–	–
37	–	$9.6\times 10^{-15}$	–	–	–	$1.2\times 10^{-2}$	–	$4.1\times 10^{-12}$
38	–	$1.9\times 10^{-17}$	–	$2.1\times 10^{-3}$	–	–	–	$1.6\times 10^{-4}$
39	–	$3.7\times 10^{-16}$	–	–	–	–	–	$3.7\times 10^{-13}$
40	–	$1.9\times 10^{-16}$	–	$2.1\times 10^{-12}$	–	–	–	$2.5\times 10^{-10}$

## Data Availability

Data are available on http://doi.org/10.6084/m9.figshare.25514854.

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
