# Peer review of "Enhancing the Efficiency of a Cybersecurity Operations Center Using Biomimetic Algorithms Empowered by Deep Q-Learning"

_biomimetics, 2024, doi:10.3390/biomimetics9060307_

Round 1
Reviewer 1 Report
Comments and Suggestions for Authors
Dear authors, congratulations for your work.
I have no relevant comments, only I suggest to make pictures bigger for a better view on paper.
Author Response
We want to thank Reviewer 1 for his suggestion, which we have applied to the manuscript.

Reviewer 2 Report
Comments and Suggestions for Authors
1、The authors presented the research question and provided a detailed showcase of the research findings, but the description of the research methodology was not sufficiently detailed.
2、It is not recommended to use the abbreviation "SOC" in the title as it can cause significant confusion among readers.
3、It is recommended that the authors present the experimental parameters and experimental platforms in a tabular format to provide detailed information.
Comments on the Quality of English LanguageMinor editing of English language required
Author Response
We want to thank Reviewer 2 for his suggestions and comments. We have created a point--to--point file with each recommendation and the improved manuscript version.

Reviewer 3 Report
Comments and Suggestions for Authors
This manuscript presents the “ Enhancing the efficiency of a Cyber SOC using biomimetic algorithms empowered by Deep Q–Learnin” and this topic of the manuscript is interesting. This manuscript outlines the demand for refined strategies in deploying Security Information and Event Management systems to support the management of Cyber Security Operations Centers amidst the complex and evolving landscape of cyber threats. The dynamic nature of these threats complicates the efficient allocation of network intrusion detection sensors, which are crucial components of robust cybersecurity frameworks. The study introduces an approach that integrates the precision of biomimetic optimization algorithms with the adaptability of Deep Q-Learning. By employing various biomimetic algorithms enhanced with deep learning, the aim is to optimize the deployment of sensors in network infrastructures while balancing network security requirements against deployment costs. Computational tests demonstrate that iterations improved through Deep Q-Learning outperform their native counterparts, highlighting the importance of reinforcement learning, specifically through Deep Q-Learning, as a powerful tool for enhancing the effectiveness of metaheuristics in addressing optimization challenges.
This manuscript is well-written. Please check as following comments and hope helpful for improvement of the manuscript. These comments can assist readers in understanding main contributions quickly.
1.It's recommended to provide more specific descriptions of the biomimetic algorithms and Deep Q-Learning methods used in the abstract to enhance readers' understanding of the research approach.
2.While the abstract mentions computational test results, it's advisable to include more concrete data here to support the reliability and persuasiveness of the conclusions.
3.There is no discussion on the cost effectiveness of the proposed method and other methods. What is the computational complexity? What is the runtime? Please include such discussions.
Author Response
We want to thank Reviewer 3 for his suggestions and comments. We have created a point--to--point file with each recommendation and the improved manuscript version.

Reviewer 4 Report
Comments and Suggestions for Authors
R. Olivares et.al., study the efficiency improvement of cyber SOC using biomometric algorithms governed by DQL. The paper was well written and scoped well. We therefore recommend the publication of this paper. Here are some minor fixes:
1) Font size is too small in Figure 3 and Figure 4 and…..
2) There is a need for more introduction to DQL's reinforcement learning and other applications.
3) I think Table 1-6 should be used as supplementary material.
4) Please check if there are any missing definitions of parameter values for the equation.
Author Response
We want to thank Reviewer 4 for his suggestions and comments. We have created a point--to--point file with each recommendation and the improved manuscript version.

Round 2
Reviewer 2 Report
Comments and Suggestions for Authors
The authors responded to my proposed modifications and suggestions very well. They have made appropriate revisions, and I believe that the requirements for public publication have been met. I agree to proceed with the publication.